

# Extraction behaviors of aqueous PEG impregnated resin system in terms of impregnation stability and recovery via protein impregnated resin interactions on bovine serum albumin

Nur Fazrin Husna Abdul Aziz[1], Sahar Abbasiliasi[2], Mazni Abu Zarin[1,3], Hui Suan Ng[4], John Chi-Wei Lan[5] and Joo Shun Tan[1]

[1] School of Industrial Technology, Universiti Sains Malaysia, Gelugor, Pulau Pinang, Malaysia
[2] Halal Products Research Institute, Universiti Putra Malaysia, UPM Serdang, Selangor, Malaysia
[3] Institute of Bioscience, Universiti Putra Malaysia, UPM Serdang, Selangor, Malaysia
[4] Faculty of Applied Sciences, UCSI University, Cheras, Kuala Lumpur, Malaysia
[5] Biorefinery and Bioprocess Engineering Laboratory, Department of Chemical Engineering and Materials Science, Yuan Ze University, Chungli, Taoyuan, Taiwan

Corresponding author
Joo Shun Tan, jooshun@usm.my

## ABSTRACT

**Background.** Current advances in biotechnology have been looked at as alternative approaches towards the limited product recovery due to time- and cost-consuming drawbacks on the conventional purification methods. This study aimed to purify bovine serum albumin (BSA) as an exemplary target product using an aqueous impregnated resin system (AIRS). This method implies the concept of hydrophobicity of polymer that impregnated into the resins and driven by electrostatic attractions and hydrophilicity of aqueous salt solution to extract the target product.

**Methods.** The extraction behaviors of impregnation in terms of stability and adsorption kinetics via protein-aqueous polymer impregnated resin were studied. Impregnation stability was determined by the leaching factor of polyethylene glycol (PEG). The major factors such as PEG molecular weights and concentration, pH of aqueous salt solution, extraction methods (sonication and agitation) and types of adsorbent material and concentration of aqueous salt phase influencing on partitioning of biomolecule were also investigated.

**Results.** For impregnation stability, the leaching factor for Amberlite XAD4 did not exceed 1%. The scanning electron microscopy (SEM) image analysis of Amberlite XAD4 attributes the structural changes with impregnation of resins. For adsorption kinetics, Freundlich adsorption isotherm with the highest $R^2$ value (0.95) gives an indication of favorable adsorption process. Performance of AIRS impregnated with 40% (w/w) of PEG 2000 was found better than aqueous-two phase system (ATPS) by yielding the highest recovery of BSA (53.72%). The outcomes of this study propound the scope for the application of AIRS in purification of biomolecules.

## INTRODUCTION

Biological products such as proteins, nucleic acids, microorganisms, animals and plant cells have been widely used in food, pharmaceutical and other industries. Since the upstream processing of these biological products is developing extensively, the downstream processing always leads to the production bottleneck. There are four stages in the downstream processing of biomolecules such as recovery, isolation, purification and polishing (*Raja et al., 2011*). Researchers have developed numerous purification strategies in order to obtain high yield of products. However, the production costs for biological products can be up to 80% (*Rosa et al., 2011*). Aqueous two-phase system (ATPS), which implies the concept of hydrophobicity and ionic charge in polymer and salt phases has been studied for primary recovery of biomolecules. However, long settling times in ATPS could be overcome by centrifugation, which consumes high energy and is considered as the drawback of this method (*Iqbal et al., 2016*). This strictly limits the development of a biomanufacturing process, especially in purification of valuable bioproducts. This drawback could be prevailed by a combination of ATPS with impregnated resin principle.

In AIRS, the aqueous polymer phase is impregnated into the solid materials while the aqueous salt phase represents the surrounding bulk phase. When the polymer impregnated solid materials are suspended in the aqueous salt phase, it creates the mass transfer between the two phases (*Tan et al., 2018*). The liquid extractant is retained within the pores of solid materials by capillary forces, which the extractant loss could be minimized compared to conventional liquid-liquid extraction. The extraction mechanism comprises of the diffusion of the solute from the surrounding aqueous bulk phase into the immobilized extractant phase. Subsequently, physical dissolving of solute in extractant takes place. A chemical complexation takes place during which extractant and solute form a complex extractant-solute. The complexation shifts the extraction equilibrium further towards the immobilized extractant phase. This method implies the concept of hydrophobicity of polymer that being impregnated into solid materials and the hydrophilicity of aqueous salt solution to extract biomolecule. Although this method has been applied in different proteins such as β-mannanase, esterase and lipase, the systemic and detailed information on this extraction behavior such as impregnation stability and adsorption kinetics have not been well-studied (*Abdul Aziz et al., 2018*; *Grilo, Raquel Aires-Barros & Azevedo, 2016*; *Tan et al., 2020*; *Tan et al., 2018*).

The aim of this study is to evaluate the extraction behaviors of aqueous polyethylene glycol (PEG) impregnated resin system in terms of impregnation stability and recovery via protein impregnated resin interactions on BSA as exemplary target product. Furthermore, the influencing factors such as molecular weight and concentration of PEG, and pH and concentration of aqueous salt phase on bovine serum albumin (BSA) partitioning behavior were examined. A comparison between aqueous PEG impregnated resin system and ATPS was evaluated. The information from the study will be used to validate the competency of the system in purification of biomolecules.

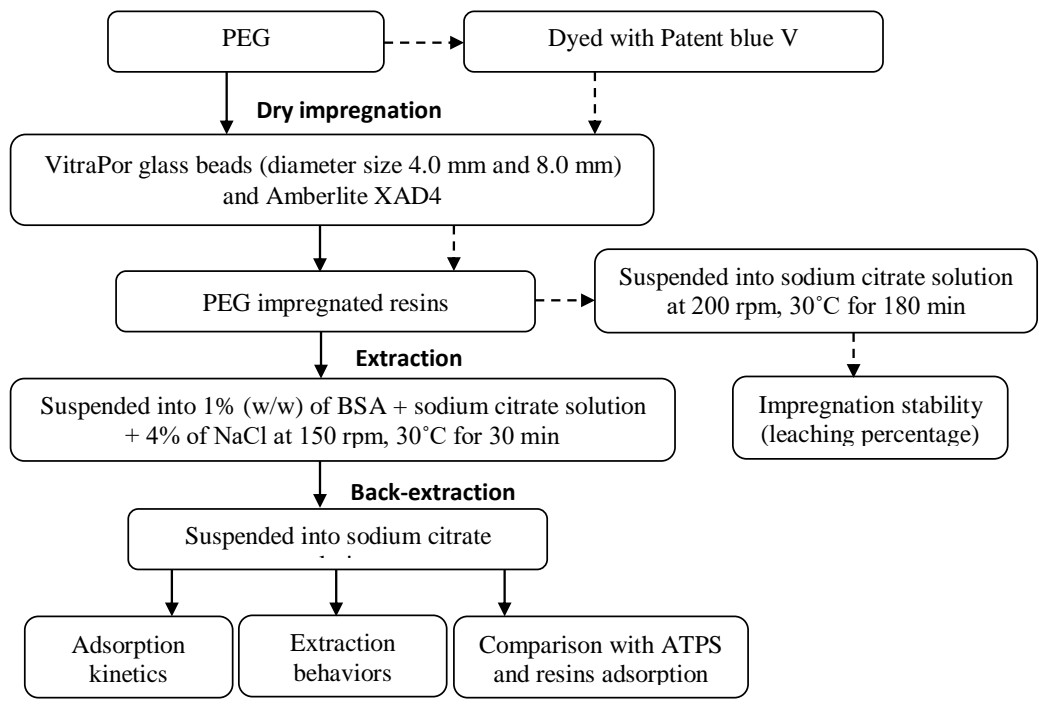

**Figure 1** Schematic overview of the experiment.

# MATERIALS AND METHODS

## Schematic overview of the experiment

The overall experimental process is presented in Fig. 1. The processes included the dry impregnation of PEG on the resins, extraction with aqueous salt phase (sodium citrate and 4% NaCl) and back-extraction prior to the analysis such as impregnation stability, adsorption kinetics, extraction behaviors and comparison with ATPS and resins adsorption.

## Chemicals and reagents

PEG with different average molecular weights of 2,000 g/mol (25322-68-3), 4,000 g/mol (25322-68-3), 6,000 g/mol (25322-68-3) and 8,000 g/mol (25322-68-3) were purchased from Milipore, Merck (Darmstadt, Germany). Potassium citrate (≥99.0%, 6100-05-6), sodium chloride (≥99.0%, 7647-14-5), BSA (>96%, 9048-46-8) and Amberlite XAD4 (20–60 mesh, 37380-42-0) were sourced from Sigma Aldrich (St. Louis, MO). The porous VitraPor glass pellets with two different particle sizes of 4.0 mm 8.00 mm were supplied from ROBU Glassfilter-Gerate GmbH (Hattert, Germany). PageBlue protein staining solution and protein loading buffer were all from Fermentas (St. Leon-Rot, Germany).

## Impregnation stability of PEG on porous solids

In order to determine the impregnation stability on the porous solids, an experiment on the impregnation stability was performed quantitatively. The experiments were started by preparing different concentrations of PEG (10% w/w, 20% w/w, 30% w/w and 40% w/w)

with different molecular weights of 2,000 g/mol, 4,000 g/mol, 6,000 g/mol and 8,000 g/mol. In order to determine the percentage of leached polymer phase, the polymer phase was dyed with 1.0 mg/mL of Patent blue V. The effects of pore size, particle size and materials on the impregnation stability were evaluated. Three types of solid materials such as VitraPor glass beads with diameter size 4.0 mm and 8.0 mm and Amberlite XAD4 (diameter size distribution of 0.25–0.84 mm) were used to impregnate the PEGs via dry impregnation method (*Van Nguyen et al., 2013*). VitraPor glass beads are inert porous glass beads with defined pore sizes of approximately 60 $\mu$m while Amberlite XAD4 is polymeric adsorbent with a pore size of approximately 0.005 $\mu$m. Both VitraPor glass beads and Amberlite XAD4 have structural rigidity, good mechanical stability and zero or minimal expansion in organic solvents. These PEG impregnated resins were suspended into 20% (w/w) of sodium citrate solution and agitated in incubator shaker (Infors HT, Switzerland) at 200 rpm, 30 °C for 180 min. The contact time was fixed at 180 min based on our preliminary study, in which the leaching percentage was stable after 30 min. For the quantitative evaluation of leaching, the leaching percentage from each sample solution at 30 min interval for 180 min was determined by measuring the absorbance of Patent blue V-dyed PEG at 620 nm using microplate reader (Halo MPR-96 Visible Microplate Reader, Dynamica). The leaching percentage of PEG were calculated using Eq. (1):

$$\text{Leaching percentage (\%)} = \frac{\text{Absorbance 620 nm of leached PEG}}{\text{Absorbance 620 nm of impregnated PEG}} \times 100. \quad (1)$$

### Experimental design

Two different aqueous solutions, PEG and sodium citrate were prepared as described by *Tan et al. (2018)*. Dry impregnation method was used to impregnate PEGs to resins. Briefly, the PEG solution was added drop by drop into the resins until all the resins were fully impregnated with PEG. A 2 g of impregnated resins was added into the aqueous salt solution containing 1% (w/w) of BSA and incubated at 30 °C for 30 min with agitation speed of 150 rpm to enhance the mass transfer rate. For back extraction, the impregnated porous solids containing the BSA were filtered out from the citrate solution containing 4% of NaCl using a sieve (100 mesh size). The excess salt solutions on the solid's surface were dried and suspended into aqueous citrate solution in the absence of NaCl for desorption process. BSA is one of the major serum proteins; it plays an important role as a result of its functional and nutritional properties which have bioactive peptides. Its low cost and wide availability compared to other proteins, its structure and functional similarity to human serum albumin enable its various biotechnological applications. Based on these aspects, BSA has been extensively applied as the model protein in various extraction and purification studies. For instance, BSA has been used as a model in the separating process using hydroxyapatite and active babassu coal (*Pereira et al., 2015*; *Ribeiro Alves et al., 2016*).

### Determination of bovine serum albumin separation performance using AIRS

To investigate the protein separation performance using AIRS, the effects of the influencing factors on partitioning behaviors were studied. The factors consisted of the PEG molecular

weights (2,000 g/mol, 4,000 g/mol, 6,000 g/mol), concentration of PEG (5% w/w, 10% w/w, 15% w/w and 20% w/w), and 8,000 g/mol), pH of sodium citrate (5.0, 6.0, 7.0, and 8.0), method of extraction *viz.* sonication and agitation, type of solid materials (VitraPor glass beads 4.0 mm diameter, VitraPor glass beads 8.0 mm and Amberlite XAD4), and salt concentration (10% w/w, 20%w/w and 30% w/w). One-factor-at-a-time was used while the other factors were fixed. The recovery yield of the protein was calculated using Eq. (2) where $C_o$ and $C_t$ referred to the initial concentration of the protein in the aqueous citrate phase and concentration of the protein in the aqueous citrate phase after back-extraction.

$$\text{Recovery yield}(\%) = \frac{Co - Ct}{Co} \times 100\% \qquad (2)$$

### Determination of protein content

Total protein concentration was determined using a Bio-Rad protein assay kit with albumin as a standard protein (*Abdul Aziz et al., 2020*). A total of 10 µL of the sample was added to 200 µL of Bradford reagent in a microtiter plate and incubated at 37 °C for 15 min. Absorbance was measured at 595 nm using a microplate reader (Halo MPR-96 Visible Microplate Reader, Dynamica).

## Statistical analysis

All the data represent the mean of three independent experiments. The errors during the measurement were indicated by standard deviations. The statistical analysis and regression coefficient ($R^2$) values of the linear form of Langmuir isotherm and Freundlich isotherm models were determined using statistical functions of Microsoft Excel 2010.

## RESULTS

### Impregnation stability

Results from the leaching percentage of different PEG molecular weights and concentration on different types of resins are presented in Table 1. The leaching factor for Amberlite XAD4 did not exceed 1%. The leaching factors for VitraPor glass beads with diameter sizes of 8.0 mm and 4.0 mm were higher as compared to XAD4. The pore size of Amberlite XAD4 (0.005 µm) is much smaller as compared to VitraPor glass beads (60 µm). The smaller pore size of the solid materials had higher impregnation stability. The PEG leaching profile depends on the PEG molecular weight and concentration, and a slower release rate is observed for the higher molecular weight and concentration. When the PEG 2000 was used, 23.74% to 20.19% of the PEG at concentrations of 10% (w/w) in the impregnation had been leached out from the VitraPor glass beads 4.0 mm and 8.0 mm, respectively. The leaching percentage decreased to 7.36% when the concentration of PEG 2000 increased to 40% (w/w). Similarly, PEG 4000 and PEG 6000 in both VitraPor glass beads had higher leaching percentages at lower concentrations (10% and 20%). The leaching percentage successfully reduced to less than 5% when the concentrations increased to 30% and 40%, respectively. Scanning electron micrographs (SEM) of Amberlite XAD4 before and after impregnation with PEG 4000 are presented in Fig. 2. Smooth surface of Amberlite XAD4 in

**Table 1 Leaching percentage of different PEG molecular weight and concentration on different types of resins.**

| Types of solid material | PEG molecular weight (g/mol) | PEG concentration (%) | Leaching factor (%) |
|---|---|---|---|
| XAD4 | 2,000 | 10 | 0.55 ± 0.02 |
| | | 20 | 0.49 ± 0.01 |
| | | 30 | 0.50 ± 0.06 |
| | | 40 | 0.54 ± 0.04 |
| | 4,000 | 10 | 0.40 ± 0.04 |
| | | 20 | 0.43 ± 0.02 |
| | | 30 | 0.50 ± 0.07 |
| | | 40 | 0.53 ± 0.03 |
| | 6,000 | 10 | 0.43 ± 0.07 |
| | | 20 | 0.45 ± 0.05 |
| | | 30 | 0.58 ± 0.01 |
| | | 40 | 0.57 ± 0.01 |
| | 8,000 | 10 | 0.49 ± 0.03 |
| | | 20 | 0.58 ± 0.04 |
| | | 30 | 0.41 ± 0.07 |
| | | 40 | 0.64 ± 0.07 |
| VitraPor 4.0 | 2,000 | 10 | 23.74 ± 0.89 |
| | | 20 | 18.74 ± 1.09 |
| | | 30 | 10.50 ± 0.87 |
| | | 40 | 7.36 ± 0.05 |
| | 4,000 | 10 | 9.02 ± 0.81 |
| | | 20 | 5.71 ± 1.07 |
| | | 30 | 4.39 ± 0.73 |
| | | 40 | 3.22 ± 0.07 |
| | 6,000 | 10 | 8.31 ± 0.01 |
| | | 20 | 6.85 ± 0.05 |
| | | 30 | 5.09 ± 0.04 |
| | | 40 | 4.06 ± 0.02 |
| | 8,000 | 10 | 6.67 ± 0.06 |
| | | 20 | 4.72 ± 0.07 |
| | | 30 | 3.70 ± 0.07 |
| | | 40 | 2.65 ± 0.01 |
| VitraPor 8.0 | 2,000 | 10 | 20.19 ± 1.08 |
| | | 20 | 17.88 ± 1.69 |
| | | 30 | 9.99 ± 1.52 |
| | | 40 | 7.07 ± 0.95 |
| | 4,000 | 10 | 10.44 ± 1.19 |
| | | 20 | 6.44 ± 0.16 |
| | | 30 | 5.05 ± 0.86 |

**Table 1** (*continued*)

| Types of solid material | PEG molecular weight (g/mol) | PEG concentration (%) | Leaching factor (%) |
|---|---|---|---|
| | | 40 | $3.73 \pm 0.07$ |
| | 6,000 | 10 | $8.99 \pm 0.43$ |
| | | 20 | $7.49 \pm 1.12$ |
| | | 30 | $5.52 \pm 0.86$ |
| | | 40 | $4.27 \pm 0.96$ |
| | 8,000 | 10 | $7.88 \pm 1.22$ |
| | | 20 | $5.05 \pm 0.02$ |
| | | 30 | $3.98 \pm 0.09$ |
| | | 40 | $3.10 \pm 0.02$ |

Fig. 2B indicated that the PEG had been impregnated into the resins, by filling the porous of the resins.

## Adsorption kinetics of BSA into resins

Comparison of different coefficients from Langmuir and Freundlich adsorption isotherm of BSA on PEG impregnated resins are presented in Table 2. Langmuir isotherm yielded a negative value of maximum BSA uptake by PEG impregnated resin, $-3.87$ mg/g while the Freundlich isotherm provided adsorption capacity, $K_f$ of 6.28 mg/g. Freundlich adsorption isotherm has the highest $R^2$ value (0.95) which is near to 1. For Freundlich model, intensity of adsorption (n) gives an indication of favorable adsorption process. This indicates the adsorption of BSA on PEG impregnated resins followed the Freundlich isotherm.

## Effect of molecular weight and concentration of PEG, pH and concentration of sodium citrate on the BSA recovery yield

Effects of the PEG molecular weight and concentration on the BSA recovery yield are presented in Fig. 3A. Four different PEG molecular weights (2,000 g/mol, 4,000 g/mol, 6,000 g/mol and 8,000 g/mol) were used, each at different concentrations (10% w/w, 20% w/w, 30% w/w and 40%w/w). The recovery of BSA increased when using lower molecular weights of PEG. PEG 2000 and PEG 4000 showed similar trend where the recovery of BSA increased as the concentration of PEG increased. In contrast, a decrease in depleted protein could be observed with higher molecular weights of PEG (6000 and 8000).

In order to determine the optimum pH for the maximum depletion of BSA, the pH of the sodium citrate solution was varied from pH 5, 6, 7 and 8 and the results are presented in Fig. 3B. According to our experiment, increasing the pH of sodium citrate solution from 5 to 7 increased the BSA yield. The yield of recovered BSA for pH 5, 6 and 7 are 30.98%, 34.97% and 51.08% respectively. However, further increase in the pH to pH 8 reduced the BSA yield (41.39%). From this experiment, pH 7.0 is selected to be used in subsequent experiment since it is the optimum pH for the BSA extraction.

The sodium citrate concentration up to 30% (w/w) was tested to determine the effect of salt concentration on the recovery of BSA in AIRS and the results are presented in Fig. 3C. The recovery of BSA was slightly different at different concentration of sodium citrate.

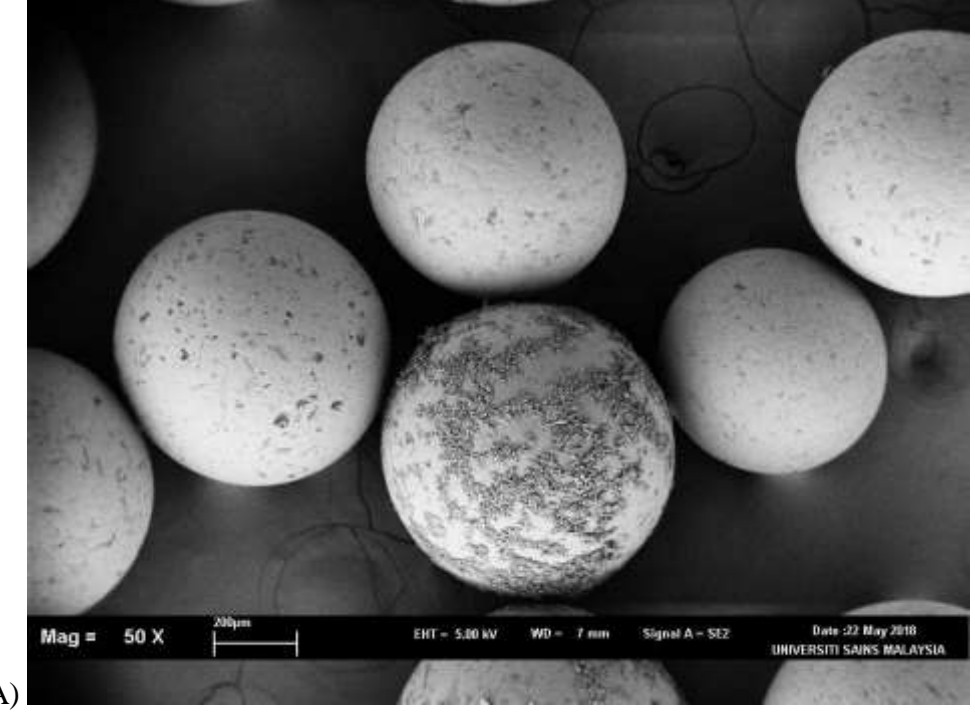

(A)

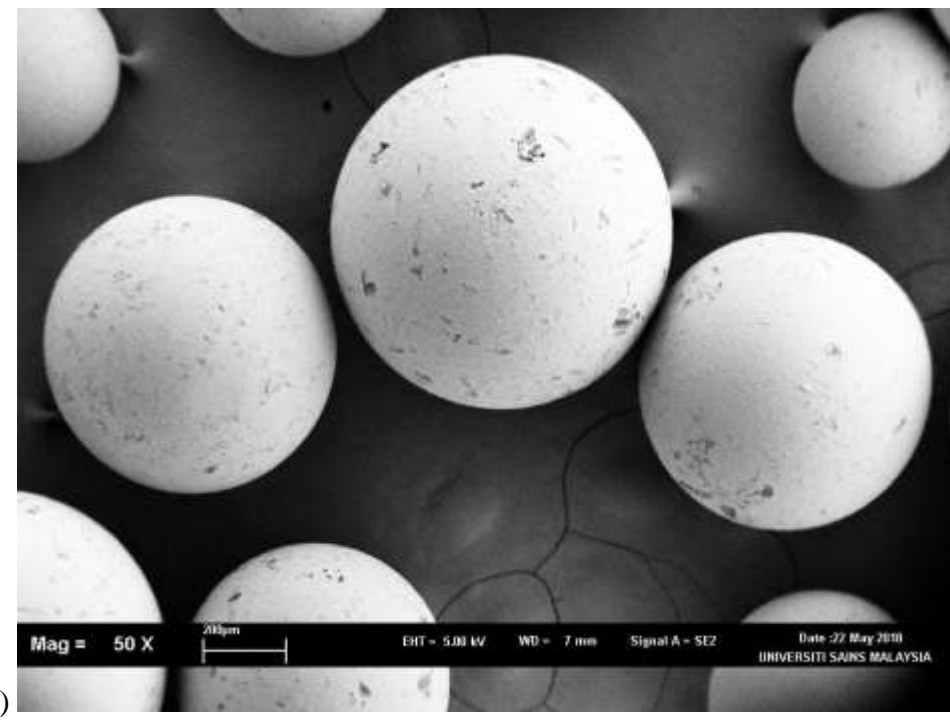

(B)

**Figure 2** SEM image analysis of Amberlite XAD4 before (A) and after (B) impregnation under 50× magnification.

**Table 2** Comparison of different coefficients from Langmuir and Freundlich adsorption isotherm of BSA on PEG impregnated resins.

| Langmuir isotherm | | | Freundlich isotherm | | |
|---|---|---|---|---|---|
| Maximum sorbate uptake, Q (mg/g) | Coefficient related to affinity between adsorbent and adsorbate, b (L/mg) | $R^2$ | Adsorption capacity, $K_f$ (mg/g) | Adsorption intensity constant, n | $R^2$ |
| −3.87 | −1.40 | 0.76 | 6.28 | 1.46 | 0.95 |

From this experiment, 10% (w/w) of sodium citrate concentration shows highest depletion of BSA (50.55%).

## Comparisons between AIRS, ATPS and adsorption method on BSA partitioning experiment

Comparison between BSA recovery yield using three different types of partitioning experiment (AIRS, ATPS and adsorption) are presented in Fig. 4. ATPS was carried out using 40% (w/w) of PEG 2000 and 10% (w/w) of sodium citrate like AIRS for comparison. AIRS yielded the highest recovery of BSA (53.72%), followed by BSA adsorbed by Amberlite XAD4 without impregnated PEG (46.38%). ATPS had the lowest BSA recovery yield (41.80%) among all the partitioning experiments. From this experiment, it was proven that by combining ATPS and adsorption method, the BSA recovery yield could be enhanced.

## DISCUSSION

The impregnation stability of different types of solid materials with different molecular weight of PEG was determined prior to BSA partitioning experiment (refer to Table 1). This is to ensure that the PEG is stably impregnated in the resins, allowing the BSA to be extracted without any leaching of PEG. This is crucial since leaching of PEG into the salt solution restrains the BSA from adsorbing to the impregnated resins. The leaching factor for Amberlite XAD4 did not exceed 1% which indicated the impregnation using Amberlite XAD4 was stable for the BSA partitioning experiment. Amberlite XAD4 is hydrophobic polyaromatic and used to adsorb small hydrophobic compounds, like PEG (*Li et al., 2001*). This chemical structure makes the PEG strongly bound into the resins, resulted in low leaching percentage. Additionally, Amberlite XAD4 represents a solid with smaller particle size distribution and smaller pore size distribution than VitraPor glass pellets. An influence of the pore size on leaching could be expected because leaching is induced at the pores outlet at the interface of the two aqueous phases. A decrease of the pore size reduces the interface and thus, potentially decreases leaching. On the other hand, smaller particle size leads to a larger interface between the phases and shorter distances for intraparticular mass transport but the polymeric nature of Amberlite XAD4 has overcome this limitation and the small particles size offers the possibility of faster partitioning of protein during purification (*Kaplanow et al., 2018*).

The results (refer to Table 1) showed that the PEG leaching was dependent on the molecular weight and concentration of PEG, with leaching percentage decreasing from low to high molecular weight and concentration and then remained below 5% of leaching

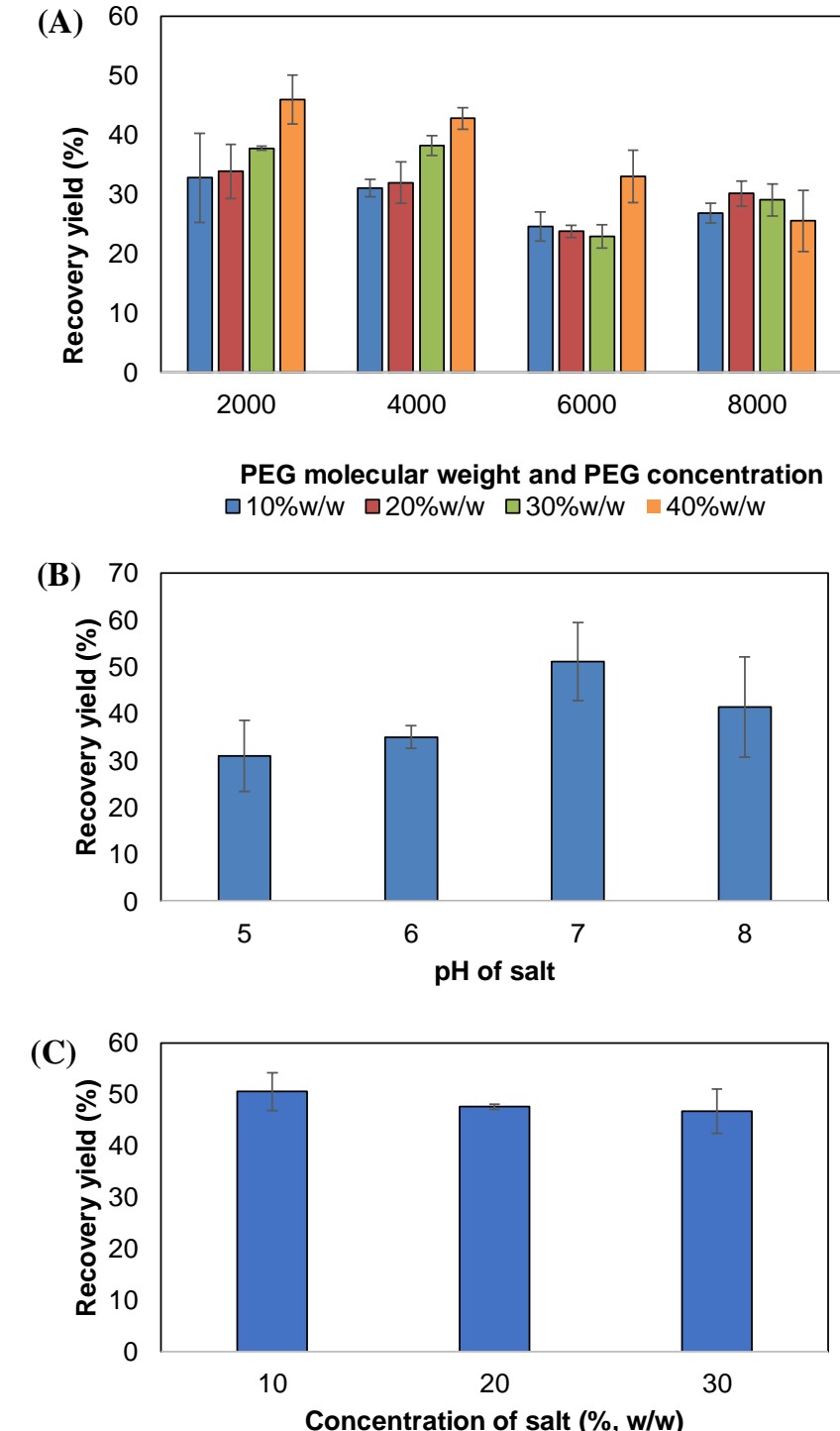

**Figure 3** **Effect of molecular weight and concentration of PEG (A), pH of sodium citrate (B) and concentration of sodium citrate (C) on BSA partitioning behavior.** The results reported as a mean of triplicate reading with an estimated error of ±5%.

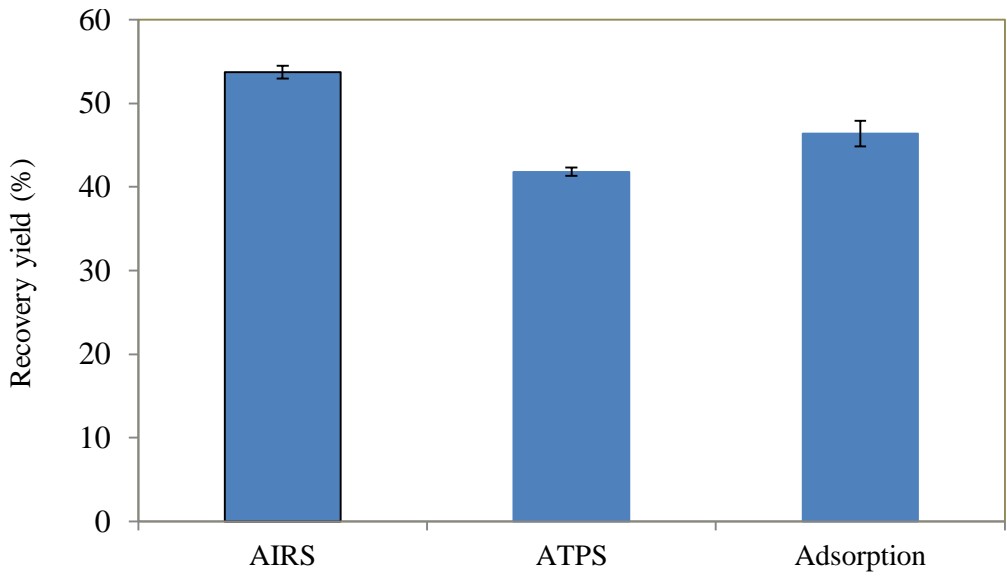

**Figure 4** **BSA recovery yield obtained using different types of partitioning experiment.** The results reported were expressed as a mean of triplicate reading with an estimated error of ±5%.

percentage for the highest molecular weight and concentration of PEG. PEG concentration at 40% (w/w) is near to saturation. This implies the existence of a maximum leaching factor, which would make it possible to fine-tune the impregnation to fit the application by selecting the optimal molecular weight and concentration of PEG. At lower concentration of PEG, a higher rate of diffusion could be observed than larger molecules as large volumes of the water has been enclosed together with the soluble PEG in the insoluble impregnated resins and thereby always gets exposed to water and prone to be dissolved in the surrounding bulk solution (*Marucci et al., 2011*). Another explanation is the differences in molecular weight of the polymers. PEG 6000 and PEG 8000 are much heavier than PEG 2000 and PEG 4000, therefore exhibits a much lower rate of diffusion. The increased molecular weight also affects the dissolution properties, a longer polymer chain is less prone to be dissolved than a shorter one (*Andersson et al., 2016*). The smooth surface of Amberlite XAD4 indicated that the PEG had been impregnated into the resins, by filling the porous of the resins (refer to Fig. 2).

In the study of adsorption kinetics, pseudo-first-order kinetic model or pseudo-second-order kinetic model are used to investigate the mechanism of solute adsorption onto extractant while adsorption isotherms such as Langmuir and Freundlich isotherms are used to describe the experimental sorption data for evaluating the adsorption equilibrium (*Murcia-Salvador et al., 2019*). To evaluate the adsorption capacity and favourable adsorption process, adsorption equilibrium is best fit for our study. The efficiency of adsorption of BSA from aqueous media using a newly developed PEG impregnated resin suggested that a contact time of 30 min sufficed to reach the saturation point and the adsorption of BSA was slowed down after that. Freundlich adsorption isotherm has the highest $R^2$ value (0.95) which is near to 1 (refer to Table 2). For Freundlich model, intensity

of adsorption (n) gives an indication of favorable adsorption process. Values of n greater than 1 represent favorable adsorption condition. This indicates the adsorption followed the Freundlich isotherm more closely than it did the Langmuir isotherm.

Since polymer is one of the major components in AIRS, it is crucial to determine the effect of the PEG molecular weight and concentration on the BSA recovery yield (refer to Fig. 3A). The recovery of BSA increased as the concentration of PEG increased. This is probably due to higher molecular weight of PEG which has higher excluded effect, thus less BSA molecules are being extracted into the PEG impregnated resins. This phenomenon results in the lower depleted BSA (*Nascimento et al., 2010*). On the other hand, higher concentration of PEG show higher depleted protein. This might be due to higher concentration of PEG which had higher impregnation stability, and caused more BSA adsorbed into the PEG. This phenomenon is in line with the findings of *Raja et al. (2011)* where higher concentration of polymer increased the differences in density, refractive index and viscosity between the polymer and aqueous salt phase. Higher density difference would prevent the polymer to be leached out from the solid materials, enables it to bind more protein molecule. Unlike tunable aqueous polymer phase impregnated resins (TAPPIR) technology presented by *Burghoff (2013)* where conventional aqueous two-phase extraction (ATPE) was relied, AIRS could impregnate maximum 40%(w/w) of impregnated polymer.

As the protein extraction in AIRS is highly affected by the charges of the protein, pH plays an important role in increasing the yield of the depleted protein (refer to Fig. 3B). As the pH of the sodium citrate solution is varied, the charge of the solute changes (*Saravanan et al., 2008*). In order to determine the optimum pH for the maximum depletion of BSA, the pH of the sodium citrate solution was varied from pH 5, 6, 7 and 8. According to our experiment, increasing the pH of sodium citrate solution from 5 to 7 increased the BSA yield. However further increase in the pH to pH 8 reduced the BSA yield. This is mainly because, the BSA becomes negatively charged as the pH of sodium citrate solution is above the isoelectric point of BSA (~5.0), the point at which the BSA has a net charge of zero (*Phan et al., 2015*). BSA was repelled by the negatively charged sodium citrate solution, resulted in the increase in adsorption of BSA and the recovery yield. In contrast, the decreasing of BSA recovery yield at pH 8 could be explained as the BSA molecular structure may have changes, causing it to unable to be partitioned into the solid material containing PEG at higher pH solution (*Silva et al., 2002*). The recovery of BSA was varied at different concentrations of sodium citrate. However, choosing the optimum salt concentration is important as it determines the maximum crude loading which it can accommodate.

A comparison on BSA recovery yield obtained using three different types of partitioning experiment (AIRS, ATPS and adsorption) in Fig. 4 showed that AIRS yielded the highest recovery of BSA (53.72%), followed by BSA adsorbed by Amberlite XAD4 without impregnated PEG (46.38%). ATPS had the lowest BSA recovery yield (41.80%) among all the partitioning experiments. From this experiment, it was proven that by combining ATPS and adsorption method, the BSA recovery yield could be enhanced.

## CONCLUSION

It could be concluded from this study that AIRS promoted a novel approach in extraction and purification of biological product. This study showed that stable impregnation could be achieved with polymeric resins and approximately 50% of BSA could be extracted using this method. This method has advantages as compare to ATPS where in this method, the stage of phase forming is omitted and only relies on the hydrophobicity of polymer and hydrophilicity and electrostatic charge of salt. It also provides wider range of polymer concentration and salt concentration. Despite the fact that this method has many advantages, the usage of resins in this experiment increase the total production cost. However, the reusability of the impregnated resin overcome this problem.

### Funding

This work was supported by the Ministry of Higher Education (MOHE), Malaysia under the Fundamental Research Grant Scheme (203/PTENKIND/6711624) and the Prototype Research Grant Scheme (203/PTENKIND/6740048). The funders had no role in study design, data collection and analysis, decision to publish, or preparation of the manuscript.

### Grant Disclosures

The following grant information was disclosed by the authors:
Ministry of Higher Education (MOHE), Malaysia: 203/PTENKIND/6711624.
Prototype Research Grant Scheme: 203/PTENKIND/6740048.

### Competing Interests

The authors declare there are no competing interests.

### Author Contributions

- Nur Fazrin Husna Abdul Aziz conceived and designed the experiments, performed the experiments, analyzed the data, prepared figures and/or tables, and approved the final draft.
- Sahar Abbasiliasi conceived and designed the experiments, analyzed the data, prepared figures and/or tables, authored or reviewed drafts of the paper, language and grammar checking, and approved the final draft.
- Mazni Abu Zarin performed the experiments, analyzed the data, prepared figures and/or tables, authored or reviewed drafts of the paper, and approved the final draft.
- Hui Suan Ng analyzed the data, authored or reviewed drafts of the paper, and approved the final draft.
- Chiwei Lan analyzed the data, authored or reviewed drafts of the paper, language and grammar checking, and approved the final draft.
- Joo Shun Tan conceived and designed the experiments, analyzed the data, prepared figures and/or tables, authored or reviewed drafts of the paper, and approved the final draft.

## Data Availability

The raw data of adsorption kinetics are available in the Supplementary File.

## Supplemental Information

Supplemental information for this article can be found online at http://dx.doi.org/10.7717/peerj.11920#supplemental-information.

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
