# Peer review of "Extraction behaviors of aqueous PEG impregnated resin system in terms of impregnation stability and recovery via protein impregnated resin interactions on bovine serum albumin"

_PeerJ, doi:10.7717/peerj.11920_

## Round 0.1 · original submission · Major Revisions

Authors, thank you for submitting your work to PeerJ. Reviewers have considered your work, with merits, but however, raised some concerns. Please, kindly attend to them carefully and diligently. Please, also provide point-by-point responses, and demonstrate the reason/reasoning behind your responses.

In addition, the Editor requests authors to consider the following:

Abstract:

In the background, it will be useful to add a sentence providing the justification of the study.

Introduction:

-Before the objective statement, why this study is being carried out appears not well justified. Kindly provide some useful literature insights regarding the principle underscoring the bioseparation of polymers. That should form the second paragraph.
-Before line 60, discuss the principle underscoring polymer impregnation by resins (or related materials). This should help make lines 64-67 make better sense.

Materials and methods:

-Please start the materials and methods with a section called 'Schematic overview of experimental program'. This should be supported by a flow diagram that explains the entire experiment, to show how the samples have been allocated, and which analyses were performed etc (two-three sentences will suffice)
-Please change 'Supplies' to Chemicals and Reagents
-Please provide a subsection captioned 'Statistical analysis'. Kindly provide details on how the data has been assembled. How mean values have been presented, how mean differences have been resolved, which platform was used to run the statistical analysis, which led to outputs of p-values, as well as R-sq values.

Results:

Line 131 -- are presented in Table 1 (please apply this same to all other places that report Tables and Figures), like Line 143,153, 173/174, etc

Discussion:

-Please, throughout the discussion, in places where the Tables and Figures are being discussed, please make sure to indicate in brackets (Refer to Table ??, or Refer to Figure ??). This is to help readers connect effectively with your work.

Overall:

Kindly make effort to attend to all the above, and provide sufficient details. This is a very promising study, and with all recommendations, the manuscript will improve. Look forward to your revised manuscript.

Reviewer 1 ·

Basic reporting

The authors report the application of AIRS for the separation of BSA. The factors influencing the performance of protein adsorption by the resin were studied systematically. As compared to ATPS and adsorption techniques, the recovery of protein by AIRS was the highest. The data and experiment design could be useful as a reference to future studies adopting AIRS as the purification method. The manuscript is recommended for publication, after consideration of the comments given below:

Experimental design

• Title: The mention of “biological product” is redundant, as BSA is commonly known as a biological product. AIRS can be mentioned in the title.
• Abbreviation. The abbreviation shall be first defined, followed by its consistent use. Please check them, e.g., AIRS, BSA, PEG. The English grammar shall be improved. A number of obvious grammar mistakes were spotted.
• Lines 62-64: The description of method is not clear. Is the salt phase responsible for biomolecule extraction? More information about AIRS can be provided in Introduction.
• Why BSA is used as an exemplary target product? Typically, BSA is the undesirable protein contaminant. Does the study intend to examine the influence of this protein in the performance AIRS? Similarly, what is the rationale behind selection of sodium citrate as the salt phase?
• Line 95: What does the ‘sample’ refer to? Solution or the impregnated resin? In Eq. 1, the absorbance of impregnated PEG is needed, but how can this be measured? Does it require the release of PEG from the resin? Besides, is 180 min sufficient to evaluate the stability of PEG impregnated on the resin?
• Line 114: What is the justification for using these 3 types of solid materials?

Validity of the findings

• Table 1: There are quite a number of interesting leaching trends exhibited by different resins. Suggest to discuss them and explain the phenomenon.
• Line 135: Is the claim on the relationship between pore size and impregnation stability based on this study? If yes, what are the pore sizes of resins used in this study?
• Line 144 and 192: Where are the results implying that 30 min of contact time is sufficient?
• Lines 206 and 207: What are the concentrations of PEG impregnated on the resin? Is there a saturation point in resin for impregnating PEG on the surface?
• Line 221: Provide reference for the pI of BSA.

Reviewer 2 ·

Basic reporting

The English language needs to be improved to ensure that an international audience can clearly understand your text.
some examples where the language could be improved include lines 48, 60, 64, 105, 111, 116, 132, 133, 148,173, 186, 202, 204, 209, 210, 211, 222 and 226.
I suggest you have a proficient English colleague and familiar with the subject matter, reviews your manuscript, or use grammar checking tools.
one example can be described as line 48, "are" need to change to "is."

Figures are appropriately Figure 2, Y axes label is missing %.

The literature references are listed without title and the end of the manuscript. Hence its is recommended to authors revise them appropriately.

Experimental design

The experimental design is well organized and appropriately addressed the aim of the study.

Validity of the findings

The results are solid, well organized and valid.

---

## Round 0.2 · Major Revisions

Please authors kindly revise your work. Look forward to your revised manuscript.

Reviewer 1 ·

Basic reporting

It is acceptable after the revision made.

Experimental design

It is acceptable after the revision made.

Validity of the findings

It is acceptable after the revision made.

Reviewer 2 ·

Basic reporting

The punctuations and grammatical errors need to be reviewed again. Please correct the list below:
Line 68 and 77, change salt aqueous solution to appropriate order as "aqueous salt solution/ phase."
Line 89 and 162, weight needs to be changed to " weights"
Line 113, please correct " Switzerland"
Line 128, the solid's surface is the correct form.
Line 164, sizes needs to be changed as " sizes"
Line 173, percentage + s = "percentages".
Line 200, following however, a comma is missing.
Line 205, slight need to be changed to " slightly"
Line 218, type +s = types.

line 147-153, Analytical methods are exactly copy-pasted from "4.5. Total Protein Content Determination" paper " Abdul Aziz, Nur Fazrin Husna, et al. "Recovery of a Bacteriocin-Like Inhibitory Substance from Lactobacillus bulgaricus FTDC 1211 Using Polyethylene-Glycol Impregnated Amberlite XAD-4 Resins System." Molecules 25.22 (2020): 5332"

Please do not copy-paste the same content and try to paraphrase it.

For lines 129-131, please indicate "Ribeiro Alves, Márcia Regina, et al. "The process of separating bovine serum albumin using hydroxyapatite and active babassu coal (Orbignya Martiana)." The Scientific World Journal 2016 (2016)." as a reference.

Experimental design

In the excel file, the model is linearized while there is no justification for such a decision. In other studies, several models of adsorption kinetics were used to fit the experimental data: the pseudo-first-order, the intraparticle diffusion model, and the pseudo-second-order kinetic model. There needs to be a justification for why the authors did not take into account other fitting models.

In the chemical and reagents part, more information regarding materials is used in the experiment. Specifying just the location of obtaining BSA does not provide the necessary information. please use this format for the chemical and reagents: (≥ 99.0%, BP671-10, Hampton, NH, USA)

Validity of the findings

If there is enough justification for why linearization instead of pseudo-first-order or pseudo-second-order is used, the linearization can be accepted. Otherwise, other fitting methods gained momentum in the literature and can be applied in this study too.

---

## Round 0.3 · accepted · Accept

Thank you authors for revising your work, and addressing all concerns. The peer-review process has been very useful, and authors have benefitted from it. The revised manuscript is now acceptable for publication. This is very useful work. Thank you for finding PeerJ as your journal of choice, and looking forward to your future scholarly contributions.

Congratulations and very best wishes.

Reviewer 2 ·

Basic reporting

It is accepted after the requested revisions are made.

Experimental design

It is accepted after the requested revisions are made.

Validity of the findings

It is accepted after the requested revisions are made.

Additional comments

It is accepted after the requested revisions are made.